# Clinical and Microbiological Features of Fulminant Haemolysis Caused by *Clostridium perfringens* Bacteraemia: Unknown Pathogenesis

**DOI:** 10.3390/microorganisms11040824

**Published:** 2023-03-23

**Authors:** Ai Suzaki, Satoshi Hayakawa

**Affiliations:** Department of Pathology and Microbiology, Nihon University School of Medicine, 30-1 Ohyaguchi Kamicho, Itabashiku, Tokyo 173-8610, Japan

**Keywords:** *Clostridium perfringens*, bacteraemia, massive intravascular haemolysis (MIH)

## Abstract

Bacteraemia brought on by Clostridium perfringens has a very low incidence but is severe and fatal in fifty per cent of cases. *C. perfringens* is a commensal anaerobic bacterium found in the environment and in the intestinal tracts of animals; it is known to produce six major toxins: α-toxin, β-toxin, ε-toxin, and others. *C. perfringens* is classified into seven types, A, B, C, D, E, F and G, according to its ability to produce α-toxin, enterotoxin, and necrotising enterotoxin. The bacterial isolates from humans include types A and F, which cause gas gangrene, hepatobiliary infection, and sepsis; massive intravascular haemolysis (MIH) occurs in 7–15% of *C. perfringens* bacteraemia cases, resulting in a rapid progression to death. We treated six patients with MIH at a single centre in Japan; however, unfortunately, they all passed away. From a clinical perspective, MIH patients tended to be younger and were more frequently male; however, there was no difference in the toxin type or genes of the bacterial isolates. In MIH cases, the level of θ-toxin in the culture supernatant of clinical isolates was proportional to the production of inflammatory cytokines in the peripheral blood, suggesting the occurrence of an intense cytokine storm. Severe and systemic haemolysis is considered an evolutionary maladaptation as it leads to the host’s death before the bacterium obtains the benefit of iron utilisation from erythrocytes. The disease’s extraordinarily quick progression and dismal prognosis necessitate a straightforward and expedient diagnosis and treatment. However, a reliable standard of diagnosis and treatment has yet to be put forward due to the lack of sufficient case analysis.

## 1. Introduction

The anaerobic bacterium *Clostridium perfringens,* widespread in the environment and gastrointestinal tract, infrequently produces bacteraemia. However, when it does occur, the fatality rate can exceed 52 per cent, making it one of the most severe types of bacteraemia [1]. *C. perfringens* infection is more lethal when massive intravascular haemolysis (MIH) develops, with fatality rates ranging from 70 to 100 per cent [2,3,4,5,6]. At Nihon University Hospital in Tokyo, we observed six cases of *C. perfringens*-related MIH.

To clarify the clinical and bacteriological profiles of MIH caused by *C. perfringens* bacteraemia, a systematic search of Pubmed over the past seven decades was conducted in this study. We searched for every English-language case report with *C. perfringens* and hemolysis (haemolysis) in the title, abstract, or main body. Then, we compared our findings to those of earlier studies.

Recent advancements in highly sensitive molecular detection techniques, such as polymerase chain reaction (PCR), have permitted the detection of bacterial genes in peripheral blood samples. After brushing or dental treatment, susceptible bacteria, such as *Polyphylomonas gingivalis*, are often detected. Nevertheless, culture tests are essential for accurately diagnosing bacteraemia, in which living bacteria are discovered in the blood. Even in developed countries, 30-day death rates range from 3 to 47% when a definitive diagnosis of clinically significant bacteraemia caused by *Staphylococcus aureus*, *Escherichia coli, Klebsiella* spp., and *Pseudomonas* spp. is made [7,8]. In most cases of bacteraemia, the bacteria are rapidly eliminated from the bloodstream; however, patients with sepsis have a poor prognosis due to systemic infection and damage to several organs. Understanding sepsis’s pathophysiology is essential, and we must investigate the specific features of relevant bacteria and host immune response.

## 2. Clostridium Perfringens

*C. perfringens* is a Gram-positive anaerobic spore-forming bacterium frequently isolated from soil and human and animal intestinal tracts. Interestingly, *C. perfringens* is also a component of the normal genital flora of 1–10% of healthy women [9]. *C. perfringens* is classified into seven types, A, B, C, D, E, F and G, according to the production of six major toxins: α-toxin, β-toxin, ε-toxin, ι-toxin, enterotoxin, and necrotising enterotoxin. All seven types produce α-toxin [10,11]. In addition to these toxins, *C. perfringens* secretes more than 20 pathogenic substances. The *C. perfringens* subtypes usually isolated from humans are type A and type F. Type A organisms only produce α-toxin (phospholipase C, CPA), which causes gas gangrene, hepatobiliary infections, sepsis [10,11], and foodborne diarrhoea [10]. Type F organisms produce CPA and enterotoxins, which cause food poisoning and nonfoodborne diarrhoea [10]. In clinical practice, bacteraemia occurs much less often than food poisoning or gas gangrene. Only approximately 0.12–0.16% of blood culture-positive samples in clinical laboratories show *Clostridium* spp., with *C. perfringens* found in 22–42% of them [12,13,14]. Clinically relevant is the fact that 7–15% of patients with *C. perfringens* bacteraemia suffer massive intravascular haemolysis (MIH) [6,15], which is characterised by the severe and systemic destruction of red blood cells. MIH is brought on by many pathogeneses [16], including immune-mediated and microangiopathic illnesses, malaria, and babesiosis [17,18]. MIH is characterised by bright red serum (Figure 1) [5,19].

Though the laboratory criteria for MIH have yet to be established, the clinical diagnosis is simple: due to the release of large amounts of haemoglobin from red blood cells into plasma, the serum of patients with MIH becomes very bright red in appearance. Among various infections that cause MIH, *C. perfringens* is one of the most critical causative organisms [1,2].

## 3. *C. perfringens* Infection with MIH

### Epidemiology

The median age of patients with *C. perfringens* bacteraemia is reported to be relatively old, ranging from 70.7 to 75.6 years [5,20,21,22]. Therefore, older age has been reported to be a risk factor for bacteraemia [1,14]. Interestingly, in our clinical cohort, the median age of bacteraemia patients with MIH was 61–66.5 years. As reported by us and others, MIH patients are suggested to be significantly younger than non-MIH patients [2,3,5]. In addition, it appears to be more prevalent in men; as reported previously, 60% [13] and 58.1% [20] were males, while the molecular basis of these gender differences is so far unknown.

*C. perfringens* bacteraemia is more prevalent in patients with diabetes, malignant neoplasms, biliary tract illness, renal failure, cirrhosis, and/or those being treated with immunosuppressive formulas [5,13,20,21]. Community-acquired infections are considered to be more common than hospital-based infections [5,13,20]. *C. perfringens*-related MIH is typically accompanied by intra-abdominal infections, liver and biliary tract infections, and lower respiratory tract infections. However, 20–30% of cases have an unknown focus [5,13,20], with no significant difference between MIH and non-MIH groups [5].

## 4. Pathogenic Factors

There have been approximately 100 case reports of *C. perfringens* bacteraemia with MIH over the past 60 years, and the number of cases has been increasing in recent years (Table 1 and Table 2). Nevertheless, because most of these cases are found in single case reports, there has been little research on causative organisms. In six cases, multiplex PCR was applied for typing toxins produced by the causative organism, all of which were type A bacteria that produce only CPA [6,21,23,24,25]. Type A *C. perfringens* bacteria, on the other hand, are common and have been linked to hepatobiliary infections, gas gangrene, and sepsis in humans. Furthermore, CPA produced by type A bacteria is produced by all types of *C. perfringens* [10]; assuming that the alpha toxin is the main pathogenic toxin in MIH is unreasonable. We typed eleven *C. perfringens* bacteraemia blood-derived clinical isolates (five from the MIH group and six from the non-MIH group) [26]. Four of the five *C. perfringens* strains that caused MIH were type A, one was type F, four of the six non-MIH strains were type A, and two were type F. Involvement of type A and F strains suggests that both may be responsible for MIH. There was no difference in the type of bacteria between the two groups.

Next, we examined the virulence factors other than the six toxins used in A-G typing for MIH. We extracted chromosomal DNA from eleven clinical isolates and compared the repertoire of known virulence-related genes between five MIH and six non-MIH groups [27]. However, there were no differences in the variation of genes considered to encode virulence factors between the groups [26].

We then compared the biological characteristics of isolates between MIH and non-MIH groups, and the growth rate of the isolates and the production of CPA did not differ [26]. In vitro human erythrocyte haemolysis experiments showed that erythrocyte haemolytic elements were present in the culture supernatants of the MIH group bacteria, with significant differences. A significant correlation was found between the erythrocyte haemolytic effect of the culture supernatant of the bacteria and the amount of θ toxin (perfringolysin O, PFO) in the culture supernatant. The amount of PFO in the culture supernatant of this clinical isolate also correlated with cytotoxicity towards human peripheral blood mononuclear cells (PBMCs) and production of interleukin-6 (IL-6) and interleukin-8 (IL-8) by human PBMCs [26]. However, there was no correlation between CPA in culture supernatants and erythrocyte haemolytic effects, cytotoxicity towards human PBMCs, or production of IL-6 and IL-8 by human PBMCs. These findings suggest that PFO is one of the most important virulence factors in *C. perfringens* bacteraemia with MIH [26].

PFO and CPA produced by *C. perfringens* have potent cytotoxic and proinflammatory cytokine-inducing effects on human blood cells [26]. PFO is produced from human PBMCs via induction of tumour necrosis factor-α (TNF-α), interferon-γ (IFN-γ), IL-2, IL-4, IL-5, IL-6, IL-7, IL-8, IL-10, IL-13, and macrophage inflammatory protein-1β (MIP-1β), inducing human erythrocyte haemolysis. PFO induces TNF-α, IL-5, IL-6, and IL-8 production more strongly than CPA. CPA induces production of IFN-γ, IL-1β, IL-2, IL-4, IL-5, IL-7, IL-8, IL-10, IL-12, IL-13, IL-17, granulocyte macrophage colony-stimulating factor (GM-CSF), and MIP-1β [26]. PFO, which induces a potent haemolytic and acute inflammatory response, plus the proinflammatory cytokine-producing effect of the CPA produced by all *C. perfringens* strains, may lead to a rapid and lethal course.

PFO is a cholesterol-dependent cytolysin (CDC) toxin family member that forms pores in membranes containing cholesterol [28]. PFO has been demonstrated to promote the development of human gas gangrene by synergistically enhancing the action of the main toxin CPA [11,29,30,31]. In animal studies, it has been documented that PFO synergistically amplifies the effects of other toxins and contributes to disease progression, such as in bovine necrohemorrhagic enteritis in combination with CPA [32] and in enterotoxaemia of sheep and goats in cooperation with ε-toxin [31]. Despite the ubiquitous occurrence of *C. perfringens* strains that produce PFO, no illnesses in which PFO has been identified as a major virulence factor have been reported [10,33,34]. More case series and research are required to establish that PFO is the primary virulence factor for MIH in *C. perfringens* bacteraemia.

## 5. Symptoms and Laboratory Findings

In MIH patients, severe primary symptoms such as altered consciousness, severe pain, shock, haematuria, and gas formation occur more frequently than in those without MIH [5]. Sudden onset of severe pain is also characteristic [5,21,23,35,36], which is difficult to distinguish from myocardial infarction or aortic dissection [35,36]. Due to the high incidence of intra-abdominal, hepatic, and biliary tract infections, significant abdominal pain is frequently observed. However, the pain may involve the entire abdomen, not just the pericardium, the right upper abdomen, the lower abdomen, or the suprapubic region. In cases of unknown aetiology, abdominal pain, chest pain, back pain, and headache could be present. Serum ALT levels are higher in those with MIH than those without MIH, despite no difference in underlying disease, foci of infection, or total bilirubin levels [5]; this may reflect a severe inflammatory response and progressive shock [37,38]. MIH patients present with tachypnoea and subsequent rapid respiratory failure, although the focus is not a respiratory infection. Initial blood gas analysis reveals acidaemia due to metabolic acidosis, followed by further hypoxaemia and respiratory acidosis, often resulting in death from acute lung injury (ALI) or acute respiratory distress syndrome (ARDS) despite ventilatory management. MIH patients may present with metabolic acidosis at an early stage, even in the absence of hypoxemia or chest X-ray abnormalities [15,23,24,35,36,39,40,41,42], and some documented patients were acidotic before the onset of intravascular haemolysis [39]. A comparison of the symptoms of *C. perfringens* sepsis with/without MIH is listed in Table 3.

High cytokine levels in the blood have been reported to rapidly cause metabolic acidosis and multiorgan failure with ARDS, acute liver failure, and acute renal failure [43]. Cytokines are also known to be induced by exotoxins of pathogenic microorganisms, such as Streptococcus pyrogenic exotoxins (SPE) [44] and *Staphylococcus aureus* toxic shock syndrome toxin [45]. Autopsy findings in reported MIH patients show simple oedema; however, without pathological findings suggestive of so-called bacterial pneumonia, such as inflammatory reactions or massive bacterial growth in the alveoli of the lungs [35,46]. These findings suggest that inflammatory cytokine levels are strongly related to the rapid progression of *C. perfringens* bacteraemia with MIH. The significantly higher age group among non-MIH patients with *C. perfringens* bacteraemia [5] may be because the production of inflammatory cytokines decreases with age [15,47]. The induction of cytokines may be related to PFO and CPA produced by *C. perfringens*. In particular, TNF-α and IL-6, which PFO strongly induces, produce pathological pain as well as fever [48], which may explain the characteristic severe pain, and it has been reported that ALI/ARDS is induced by IL-6, IL-8, and IL-10 [49,50]. However, because the sera from patients with MIH were strongly haemolytic, it was difficult to measure cytokine levels or *C. perfringens*-produced toxins in serum samples using ELISA or other laboratory methods. This was also the case at other centres, which may have hindered pathophysiologic analysis. 

## 6. Diagnosis with/without Bacterial Culture

Among patients with *C. perfringens* bacteraemia, 40–55% have polymicrobial bacteraemia [5,20], with no difference between the MIH and non-MIH groups. Regarding susceptibility to antimicrobial agents, both groups were found to be sensitive to penicillin and carbapenem, while susceptibility to clindamycin tended to be lower in the MIH group [5]. Blood culture tests for *C. perfringens* provide positive results in an average of 16.9 hours, which is faster than those for other Clostridium species [13]. This is owing to a doubling period of around 7 minutes, which is significantly faster than other bacteria. Due to the rapid progression of MIH, however, patients cannot be treated based on culture results, and many perish by the time the results are acquired. Gram staining of the buffy coat and the presence of Gram-positive rods will lead to an early diagnosis if MIH is suspected based on patient serum results (Figure 2) [5,51].

## 7. State-of-the-Art Treatment and Prognosis 

*C. perfringens* bacteraemia with MIH has a poor prognosis. Because reported cases suggested that life expectancy was significantly lower among MIH patients, with attributed mortality of 6/6, or 100%, compared to 13/54, or 24.1%, among non-MIH patients (*p* 0.001), the mean time between bacteraemia and death was 0.18 days (range: 0.04–1.08 days) in MIH versus 32 days (5–73 days) in controls (*p* 0.001).

Therefore, it is strongly recommended that a clinician who encounters such cases begin potent systemic antibacterial therapy as soon as it is suspected. Although there are few reports, some suggest that penicillin plus clindamycin lowers the risk of death compared to penicillin alone or other antimicrobial agents [15]. In association with antimicrobial therapy, surgical resection of infected lesions has been reported to improve survival [3] significantly. However, patients with *C. perfringens* bacteraemia with MIH are already in a state of shock when the disease is suspected. Many patients die before they can benefit from treatment [2,33], making the choice of surgical intervention difficult. Patients who can undergo surgery may have MIH but are relatively haemodynamically stable and have a high chance of survival [3]. Blood purification [4] and hyperbaric oxygen therapy (HBOT) [3] have also been attempted. However, the prognosis for *C. perfringens* bacteraemia with MIH is extremely poor, with a median time from admission to death of only 10 hours, despite intensive care [2,3,5]. Even when apparently susceptible antimicrobial agents are used, mortality rates range from 70 to 100% [2,3,4,5,6], and there has been no decrease in mortality over the past 30 years. So far, antimicrobials have been used to treat cases of other Clostridium pathogens, including *C. difficile* infection (CDI), while *C. tetani* and *C. botulinum* infections are already effectively treated with antitoxins. [52]. Antitoxin therapy has also been utilised to treat gas gangrene induced by *C. perfringens* [53]. In newborn piglets infected with *C. perfringens* type C, commercial swine anti-beta toxoid vaccinations have also been reported to be efficacious against necrotising enterocolitis [54]. To treat *C. perfringens* bacteremia with MIH, we suggest using anti-PFO toxin therapy and establishing cytokine-targeted treatment with anti-IL-6 antibodies [34]. We suggest this because anti-IL-6 monoclonal antibody medicines were initially developed to treat persistent inflammation, such as that caused by autoimmune disorders. Furthermore, multiple data suggest that they are efficacious for COVID-19-induced cytokine storms.

## 8. Inflammatory Foci or Bacterial Translocation Preceding *C. pefringens* Invasion Pathways

Bacterial translocation or inflammatory foci can come before *C. pefringens* invasion routes. Numerous anaerobic bacteria are present in the commensal flora on the skin, oral cavity, gastrointestinal tract, and vagina. Anaerobic bacteria can infect wounded tissues even in the presence of a normally functioning host immune system. In some instances, the illness is caused by a combination of aerobic and anaerobic bacteria, as opposed to anaerobic bacteria alone. The main focus of mixed infection is necrotic tissue resulting from trauma, ischaemia, or malignancies. Neoangiogenesis is induced by inflamed tissue, which boosts blood flow and makes it possible for aerobic or anaerobic bacteria to enter the bloodstream and cause bacteremia. As a result, after an infection has established itself at the primary site, it may spread to other areas through the bloodstream and induce systemic effects, such as disseminated intravascular coagulation (DIC), cytokine storms, and MIH. Most anaerobic infections do not result in DIC when bacteraemia occurs. However, clostridial infections can infrequently result in coagulopathy related to sepsis. In 20–30% of the 60 cases of *C. perfringens* bacteraemia we documented, the main inflammatory lesion of the bacterial entrance was difficult to identify. This result is reflected in reports from other centres.

This means that *C. perfringens* can enter the bloodstream without necessitating the formation of inflammatory foci anywhere in the body, which could have catastrophic effects. In such circumstances, the most likely entrance route is bacterial translocation from the gastrointestinal system. Small amounts of enteric bacteria can enter the bloodstream even in healthy people; however, they have been detoxified in the liver via the portal vein. The reticuloendothelial system also processes them at the spleen. However, for unknown reasons, such as an abnormality in the intestinal microbiota or a disruption in the intestinal mucosal barrier, more bacteria are allowed to enter into circulation. Inadequate processing of blood bacteria causes systemic bacteraemia. Furthermore, as previously reported, *C. perfringens*-produced cholesterol-dependent cytolysin (CDC) is another candidate mucosal disruptor [55]. This bacterial species damages the mucosal function to prevent bacterial entry into the systemic circulation by silencing mucosal macrophages that protect the intestinal barrier function.

## 9. Evolutionary Significance of Bacterial Haemolysis

Haemolysis is the breakdown of red blood cells and derives from the Greek word αιμόλυση, meaning “destruction of blood.” It emerged in evolution because of the use of host animals with red blood cells as a source of nutrients. Iron is an essential component for bacterial growth, and it is believed that the breakdown of erythrocytes, which contain large amounts of iron, promotes bacterial growth. Many human infections, particularly Gram-positive cocci, produce haemolysin, a substance that induces haemolysis. In clinical bacteriology, bacteria can also be categorised based on their haemolysis pattern. The haemolytic pattern of bacterial colonies grown on blood agar media determines whether they cause alpha- or beta-haemolysis. Alpha-haemolysis, in which hydrogen peroxide produced by the bacteria oxidises haemoglobin to become methaemoglobin, a green oxidised derivative, is indicative of *Streptococcus pneumoniae* and *Streptococcus viridans*. Group A streptococci (GAS) and *Streptococcus dysgalactae* produce beta-haemolysis (complete haemolysis), a condition in which red blood cells in the medium around and under the colony fully decompose and become transparent. Streptolysin O (SLO) and streptolysin S (SLS), both of which are generated by bacteria, are responsible. SLS specifically damages immunological cells, including polymorphonuclear leukocytes and lymphocytes. For convenience, bacteria that do not cause haemolysis are referred to as “gamma-haemolytic”, which include *Enterococcus faecalis* and *Staphylococcus epidermidis*, commensal bacteria of the gastrointestinal system and skin that have no direct contact with red blood cells. *C. perfringens* is a common bacterium in the gastrointestinal tract and on the skin. As mentioned previously, *C. perfringens* is found in the environment, intestines, and vagina. It rarely interacts with erythrocytes. Therefore, it is unlikely that *C. perfringens* actively uses the iron in erythrocytes released by haemolysis, and the damaging haemolysis that occurs with bacteraemia is either incidental or an overreaction by the host. We believe that the haemolysis associated with bacteraemia is either an evolutionary accident or an overreaction by the host, as killing the host would render the parasite bacteria unable to thrive.

## 10. Involvement of Haemolysis in Pathophysiology of *C*. *Perfringens* MIH and Potential for Iron/Haem Scavenging Therapy

Haemolysis is caused by various factors; however, there is growing evidence that free haem and iron can harm the body by activating endothelial and immune cells [56]. The haem and iron released from erythrocytes as a result of haemolysis promote leukocyte adhesion to endothelial cells, causing damage not only to blood vessels but also to the various functions of systemic organs. Increased circulating free Hb concentrations reduce the Hb scavenger haptoglobin (Hp), part of a critical detoxification system in mammals, scavenging haemolytic by-products in the blood and maintaining normal intracellular metabolism [57,58]. Furthermore, the haemopexin (Hx) scavenger and the intracellular enzymes haem oxygenase (HO-1 and -2) are involved [59]. Hp and Hx bind to Hb and haem with high affinity and transport Hb to macrophages and haem to hepatocytes, respectively, preventing oxidative enhancement in the circulation and non-specific uptake in non-target cells; HO degrades haemo porphyrins to iron-carbon monoxide and biliverdin, which have anti-inflammatory, anti-oxidative, and anti-apoptotic effects. Released iron, on the other hand, forms ferritin-heavy chains (H-ferritin). It is oxidised to ferrous iron (Fe^2+^) by ferroxidase activity; if the haem detoxification system is saturated with high levels of haemolysis, these systems fail [60]. Increased oxidative stress and elevated levels of soluble vascular cell adhesion molecule-1 (sVCAM-1), soluble endothelial selectin (sE-selectin), tumour necrosis factor (TNF), interleukin-6 (IL-6), and vascular endothelial growth factor (VEGF) result from scavenger depletion [61]. Soluble haem is a potent inducer of type I IFN, which causes haemophagocytic syndrome and worsens the patient’s prognosis. These findings suggest that *C. perfringens*-caused severe haemolysis has a common aetiology, with non-infectious haemolysis as a cause of death in patients. In this context, we advocate for early iron/haem scavenging therapy in non-infectious haemolytic diseases.

## 11. Local or Systemic Modulation of Immune Function by Genus Clostridium and How to Control Them

Bacteria of the genus Clostridium are known for their strong immunoregulatory effects. Regulatory T cells, essential for maintaining pregnancy and preventing the onset of autoimmune diseases, are induced by bacteria of the genus Clostridium living in the intestinal tract [62]. The number of Treg cells in the large intestines of mice raised in a normal environment is drastically higher than in the intestines of mice raised in a sterile environment (aseptic mice). When sterile mice are inoculated with various intestinal bacteria, Clostridium spp. markedly increase the number of Treg cells in the colon. It has long been known that mice with high levels of Clostridium spp. are less prone to develop enteritis and allergic reactions; however, when the intestinal bacteria are eradicated with antimicrobial agents, the incidence of these reactions increases. Similarly, in humans, Clostridia in the intestinal tract are supposed to induce regulatory T cells. In our study of the intestinal microbiota in children, we reported that intestinal Clostridia are closely related to the development of orthostatic dysregulation and allergic diseases [63]. It is thought that there are two types of clostridia: so-called “good” clostridia, which are useful for our intestinal environment and immune function, and “bad” clostridia, which are the focus of inflammation and induce excessive immunosuppression. It is difficult to speculate whether the bias in the intestinal microflora can be corrected simply by the administration of probiotics or antibiotics. The most probable candidate is breastfeeding during the neonatal period. This is because *C. perfringens* colonisation occurs early after birth and persists for an extended period of time throughout life but is reported to be more frequent in children born by caesarean section [64]. Breastfeeding is strongly recommended to maintain a healthy gut microbiota throughout life [65] because it effectively prevents necrotising enterocolitis caused by *C. perfringens* in infants born by either vaginal delivery or caesarean section.

## 12. Conclusions

Bacteraemia associated with MIH advances rapidly, and patients with suspected cases frequently die before blood cultures can be completed because they are in serious shock condition when they arrive at the hospital. The clinical features of *C. perfringens* bacteraemia are severe pain at onset, impaired consciousness, shock, haematuria, metabolic acidosis, and gas formation. When a blood sample from a person with these symptoms shows intravascular haemolysis, the physician should be ready for a very fulminant case outcome. Future multicentre, case-intensive clinical studies using, if possible, the prospective approach is desirable to elucidate the pathophysiology of this rare but fatal disease and to establish treatment for it. 

In addition, the identification of the molecular backgrounds of the MIH-causing *C. perfringens* substrains is required. 

There have been numerous instances of various Clostridium species causing haemolysis [3]. However, this is uncommon, indicating that MIH-related *C. perfringens* may have a particular mechanism. Moreover, *C. perfringens* occasionally inhabits the digestive and vaginal tracts. While the number of clinical cases in humans is exceedingly low, it may be possible to clarify the aetiology of MIH by using animal models to identify the strains prone to cause MIH, the relevant genes, and the host immune response and cytokine patterns.

## Figures and Tables

**Figure 1 microorganisms-11-00824-f001:**
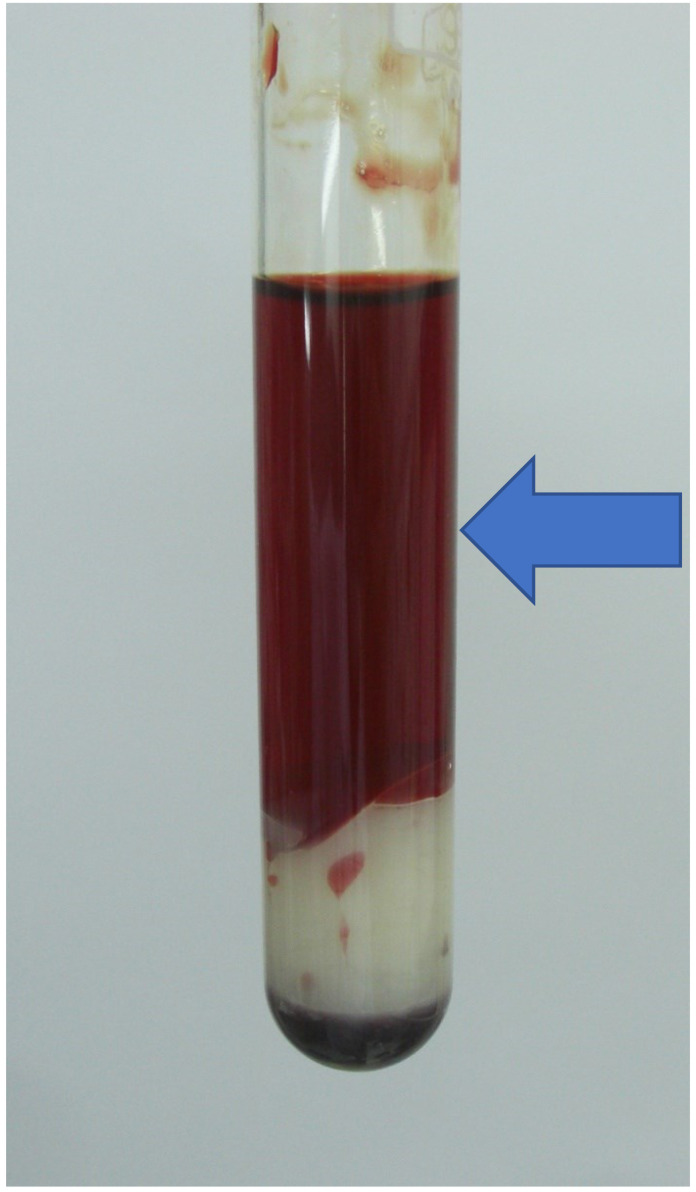
Macroscopic appearance of massive haemolysis caused by *C. perfringens* bacteraemia. Bright red appearance of the serum (arrow).

**Figure 2 microorganisms-11-00824-f002:**
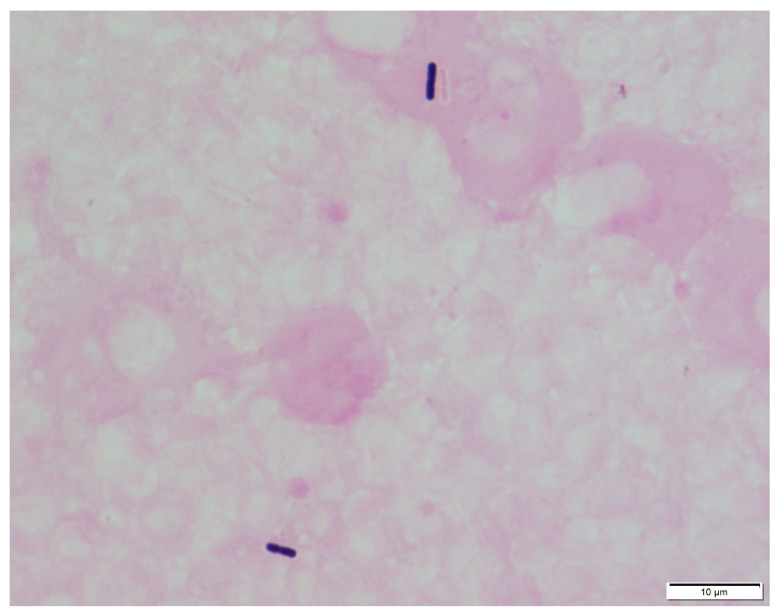
Microscopic appearance of *C. perfringens* prepared from the buffy coat of a patient’s peripheral blood and Gram staining (×1000).

**Table 1 microorganisms-11-00824-t001:** Reported case numbers of *C. perfringens* bacteraemia with MIH from 1951 to 2022.

By Decade	Reported Cases
1951–1960	3
1961–1970	2
1971–1980	2
1981–1990	8
1991–2000	15
2001–2010	30
2011–2020	37
2021–2022	18
Total	115

**Table 2 microorganisms-11-00824-t002:** All reported cases of *C. perfringens* bacteraemia with MIH from 1991 to 2022.

	Author	Year	Age	Sex	Origin Infection	Positive Culture	Survival	Toxin		Toxinotype
								CPA	CPE	
1	Bätge	1992	61	M	Liver ab-scess	Blood	Yes	NR	NR	
2	Ifthikaruddin	1992	54	F	Unknown	Blood	No	NR	NR	
3	Hübl	1993	84	F	Intestinal	Blood	No	+	NR	
4	Rogstad	1993	61	M	Micro liver abscess	Blood, liver	No	NR	NR	
5	Clarke	1994	53	F	Necrotising enteritis	Blood, peritoneal fluid	Yes	NR	NR	
6	Meyerhoff	1995	66	F	Unknown	Blood	No	NR	NR	
7	Gutiérrez	1995	74	M	Micro liver abscess	Blood	No	NR	NR	
8	Singh	1996	73	F	Unknown	Blood	No	NR	NR	
9	Bush	1996	58	F	Biliary	Blood	Yes	NR	NR	
10	Jones	1996	66	F	Liver abscess	Blood, liver abscess	No	NR	NR	
11	Pun	1996	47	M	Cholangitis	Blood	No	NR	NR	
12	Singer	1997	55	F	Unknown	Blood	No	NR	NR	
13	Alvarez	1999	77	F	Abdominal	Blood	No	NR	NR	
14	Thomas	1999	73	M	Cholecystitis	Blood	Yes	NR	NR	
15	Eckel	2000	65	F	Liver abscess	Blood	Yes	NR	NR	
16	Kreidl	2002	80	M	Liver abscess	Blood, liver abscess	No	NR	NR	
17	Barrett	2002	NR	F	Septic abortion	Blood	No	NR	NR	
18	Jimenez	2002	79	M	Unknown	Blood	No	NR	NR	
19	Halpin	2002	29	F	Post-caesarean endometritis	Blood	Yes	NR	NR	
20	Hamoda	2002	39	F	Post-amniocentesis endometritis	Blood	Yes	NR	NR	
21	Ikegami	2004	67	M	Acute pancreatitis	Pancreas	Yes	NR	NR	
22	Vaiopoulos	2004	74	M	Intestinal and biliary	Blood	No	NR	NR	
23	Solis	2004	50	M	Hepatic gas gangrene	Donor liver	No	NR	NR	
24	Rodriguez	2005	57	M	Biliary	Blood	No	NR	NR	
25	Pirrotta	2005	50	M	Unknown	Blood, stool	No	NR	NR	
26	Au	2005	65	M	Liver abscess	NR	No	NR	NR	
27	McArthur	2006	49	M	Abdominal	Blood	No	NR	NR	
28	Daly	2006	80	M	Liver abscess	Blood	No	NR	NR	
29	Kwon	2006	71	F	Unknown	Blood	No	NR	NR	
30	Loran	2006	69	F	Liver abscess	NR	No	NR	NR	
31	Ohtani	2006	78	M	Liver abscess	Blood, liver abscess	No	NR	NR	
32	Eigneberger	2006	60	M	Liver abscess	Liver (Gram stain)	No	NR	NR	
33	Poulou	2007	74	M	Unknown	Blood	No	Lecithinase	NR	
34	Kapoor	2007	58	M	Unknown	Blood	No	NR	NR	
35	Poon	2007	64	F	Hepatobiliary	Blood	No	NR	NR	
36	Nadisauskiene	2008	31	F	Post-caesarean endometritis	Blood	No	NR	NR	
37	Egyed	2008	39	F	Unknown	Blood	Yes	NR	NR	
38	Hess	2008	81	M	Diverticulitis	Blood, brain, heart, spleen	No	NR	NR	
39	Boyd	2009	46	M	Cholecystitis	Blood	No	NR	NR	
40	Uppal	2009	61	M	Unknown	Blood	No	NR	NR	
41	Merino	2010	83	F	Liver abscess	Blood	No	NR	NR	
42	Ng	2010	61	F	Liver abscess	Blood	Yes	NR	NR	
43	Rajendran	2010	58	M	Liver abscess	Blood, liver abscess, gall bladder	Yes	NR	NR	
44	Bunderen	2010	74	M	Cholangitis	Blood	Yes	NR	NR	
45	Bryant	2010	60	F	Uterus	Blood, intrauterine	Yes	NR	NR	
46	Stroumsa	2011	41	F	Uterine myoma	Blood	Yes	NR	NR	
47	Qandeel	2012	59	M	Liver abscess (post-laparoscopic cholecystectomy)	Blood	Yes	NR	NR	
48	Watt	2012	52	M	Pan-enteritis	Blood	Yes	NR	NR	
49	Law	2012	50	F	Liver abscess	Blood	No	NR	NR	
50	Okon	2013	71	M	Unknown	Blood, CSF	No	NR	NR	
51	Cécilia	2013	64	M	Unknown	Blood	No	NR	NR	
52	Dutton	2013	66	M	NR	Blood	No	NR	NR	
53	Kitterer	2014	71	M	Liver abscess	Blood	No	NR	NR	
54	Kurasawa	2014	65	M	Liver abscess	Blood	No	NR	NR	
55	Renaudon-Smith	2014	37	M	Liver abscess	Blood	Yes	NR	NR	
56	Simon	2014	79	F	Unknown	Blood	No	NR	NR	
57	Shindo	2015	73	F	Liver abscess	Liver abscess	No	+	−	A
58	Khan	2015	77	M	Cholecystitis, liver abscess	Liver (Gram stain)	No	NR	NR	
59	Cochrane	2015	65	F	Emphysematous cholecystitis	Blood	Yes	NR	NR	
60	Yamaguchi	2015	80–89	F	Unknown	Bile, pleural effusions	No	NR	NR	
61	Li	2015	71	M	Liver abscess (post-TACE)	Blood	Yes	NR	NR	
62	Medrano-Juarez	2016	32	M	Unknown	Blood	Yes	NR	NR	
63	Lim	2016	58	M	Liver abscess	Blood	No	NR	NR	
64	Hashiba	2016	82	M	Liver abscess, emphysematous cholecystitis	Blood	No	+	−	A
65	Sarvari	2016	76	F	Emphysematous gastritis	Intestine subcutaneous tissue	No	NR	NR	
66	Carretero	2016	65	M	Liver abscess	Blood, liver abscess	Yes	NR	NR	
67	Kent	2017	74	F	Enteritis	Blood	No	NR	NR	
68	Kukul	2017	17	M	Gastrointestinal tract	Quadratus muscle	No	NR	NR	
69	Balan	2017	71	F	Unknown	Blood	No	NR	NR	
70	Ewing	2017	53	F	Necrotising fasciitis	Wound	No	NR	NR	
71	Shibazaki	2018	68	F	Liver abscess	Blood	No	NR	NR	
72	Wild	2018	81	F	Unknown	Blood	No	+	−	A
73	Gelonch	2018	66	M	Liver abscess	NR	No	NR	NR	
74	Gelonch	2018	63	M	Liver abscess	NR	No	NR	NR	
75	Uojima	2019	83	M	Liver abscess (post-TACE)	Liver abscess	No	NR	NR	
76	Sakaue	2019	76	M	Liver abscess	Blood	No	+	−	A
77	Kawakami	2020	83	M	Pelvic abscess	Blood, intraabdominal samples	No	NR	NR	
78	Fujikawa	2020	77	F	Liver abscess	Blood	No	NR	NR	
79	Chinen	2020	80	F	Liver abscess	Blood, liver abscess	No	NR	NR	
80	Smit	2020	61	M	Liver abscess	Blood	No	NR	NR	
81	Smit	2020	71	F	Unknown	Blood	No	NR	NR	
82	Koubaissi	2020	50	M	Abdominal	Blood	No	NR	NR	
83	Poletti	2021	64	F	Unknown	Blood	No	NR	NR	
84	Liu	2021	21	M	Intestine	Blood	Yes	NR	NR	
85	Liu	2021	42	M	Intestine	Blood	No	NR	NR	
86	Fukui	2021	69	M	Unknown	Blood	No	+	−	A
87	Olds	2021	85	F	Liver abscess	Blood	No	NR	NR	
88	Bibi	2021	77	M	Cholecystitis (Post ERCP)	Blood	No	NR	NR	
89	Guo	2021	62	M	Hepatoma (post-microwave ablation)	Blood	No	NR	NR	
90	Woittiez	2021	65	M	Gangrenous cholecystitis	Blood, liver abscess	No	+	−	A
91	Woittiez	2021	69	M	Liver meta? (post-microwave ablation)	Blood	No	+	−	A
92	Takahashi	2022	70	M	Liver abscess	Blood, liver abscess	Yes	NR	NR	
93	Wong	2022	80	M	Liver abscess	Blood	Yes	NR	NR	
94	Kohya	2022	60	M	Perforation in ascending colon cancer	Blood	No	+	+	F
95	Suzaki	2022	69	M	Enteritis	Blood	No	+	−	A
96	Suzaki	2022	65	F	Cholecystitis	Blood	No	+	−	A
97	Suzaki	2022	68	F	Ovarian tumour	Blood	No	+	−	A
98	Suzaki	2022	46	M	Trauma	Blood	No	+	+	F
99	Suzaki	2022	72	M	Unknown	Blood	No	+	−	A
100	Suzaki	2022	58	M	Unknown	Blood	No	+	−	A

CPA: phospholipase C; CPE: *Clostridium perfringens* enterotoxin; NR: not reported; CSF: cerebrospinal fluid; TACE: transarterial chemo-embolisation; ERCP: endoscopic retrograde cholangiopancreatography.

**Table 3 microorganisms-11-00824-t003:** Clinical profiles of *C.perfringens* infection with/without MIH.

	MIH (n = 6)	w/o MIH (n = 54)	
Median age	66.5 (46–72 years)	77.0 (46–72 years)	*p* = 0.017
Loss of consciousness	6/6 (100%)	19/54 (35.2%)	*p* = 0.004
Severe pain at the onset	4/6 (66.7%)	10/54 (18.5%)	*p* = 0.008
Shock at onset	3/6 (50%)	3/54 (5.6%)	*p* = 0.010
Haematuria	2/6 (33.3%)	1/54 (1.9%)	*p* = 0.024
GAS formation	3/6 (50%)	4/54 (7.4%)	*p* = 0.017

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
