# Peer review of "Clinical and Microbiological Features of Fulminant Haemolysis Caused by Clostridium perfringens Bacteraemia: Unknown Pathogenesis"

_microorganisms, 2023, doi:10.3390/microorganisms11040824_

Round 1

Reviewer 1 Report

It is a well-written and very interesting review of C.perfrigens sepsis and MIH.

I would like to suggest minor comments:

1. Please, define MIH (laboratorio criteria).

2. In "Symptoms and laboratory" paragraph, please explain where "severe pain" is located.

Furthermore, last sentence in the first paragraph is not exacto. It is possibile yo have metabolic acidosis without lung disease. Metabolic acidosis is probable due to lactic acydosis because of sepsis. Please, reformule this sentence. 

thank you

Author Response

Our response to the referee’s comments

Referee 1

It is a well-written and very interesting review of C.perfrigens sepsis and MIH.

Response: Thank you for your favorable comments.

I would like to suggest minor comments:

  1. Please, define MIH (laboratorio criteria).

Response: Thank you for your comment.  We looked through many haematology and infectious diseases textbooks as well as medical articles searchable on Pubmed. However, none of them defined laboratory criteria for MIH caused by a C. perfringens infection. We have therefore stated this fact and added the following.

(P ï¼– L8-11  ) Though the laboratory criteria for MIH have yet to be established, the clinical diagnosis is simple: by releasing large amounts of haemoglobin from red blood cells into plasma the serum of patients with MIH becomes a very bright red appearance.

In "Symptoms and laboratory" paragraph, please explain where "severe pain" is located.

Response:  We really appreciate your comments. We have added as follows.

(P10 L20 - P11L1 ) - Due to the high incidence of intra-abdominal, hepatic, and biliary tract infections, significant abdominal pain is frequently observed. However, the pain may involve the entire abdomen, not just the pericardium, the right upper abdomen, the lower abdomen, or the suprapubic region. In cases of unknown aetiology, abdominal pain, chest pain, back pain, and headache could be present.

Furthermore, last sentence in the first paragraph is not exacto. It is possibile yo have metabolic acidosis without lung disease. Metabolic acidosis is probable due to lactic acydosis because of sepsis. Please, reformule this sentence. 

Response: Thank you for your comment. We have revised it as follows.

(P11  L11-L16  )   MIH patients may present with metabolic acidosis at an early stage, even in the absence of hypoxemia or chest X-ray abnormalities [15,23,24,35,36,39-42], and some have been documented patients were acidotic before the onset of intravascular haemolysis [39]. A comparison of the symptoms of C.perfringes sepsis with/without MIH is listed in Table 3.

Reviewer 2 Report

General comments

=============

I appreciated the opportunity to peer-review your work on the review for Clostridium perfringens bacteremia. The authors should elaborate the overall structure. It is important for the manuscript to have a clear and logical flow of information to help readers understand the research question, methods used, and findings obtained. The authors can consider providing a brief outline or summary of the manuscript in the introduction section, which can help readers get a better understanding of the structure of the manuscript.

Specific comments

=============

Major comments

---------------------

1. The major consideration is that the authors needed to clarify the review part and the authors’ investigation part. 

2. The authors should provide more information on the methods they used to conduct the review of existing literature. To do this, the authors can consider including a detailed description of the search strategy, including the databases used and the keywords searched.

3. Please clarify the content for Clostridium perfringens with/without MIH and other Clostridium pathogens. Please clarify the treatment for adult and newborn, especially for the treatment part.

4. Please add the age of non-MIH patients and statistical significance before the following sentences: “Therefore, our study shows that MIH patients are significantly younger than non-MIH patients.” Same as the following sentences: “In MIH patients, severe primary symptoms such as altered consciousness, severe pain, shock, haematuria, and gas formation occur more frequently than in those without

MIH.”

5. Please add the prevalence of males to show more common.

6. Please add the table to compare the background, serotype, clinical presentation, and laboratory data between C. perfringens with MIH and that without MIH from 106 cases.

7. Please add the poor prognosis as a number.

Minor comments

---------------------

6. Please clarify the legend in table 1. It was doubled.

7. Please elaborate on table 1.

8. Please add the arrow(s) in figure 1 to clarify the massive hemolysis for unfamiliar readers.

Author Response

Referee2

Comments and Suggestions for Authors

General comments

=============

I appreciated the opportunity to peer-review your work on the review for Clostridium perfringens bacteremia. The authors should elaborate the overall structure. It is important for the manuscript to have a clear and logical flow of information to help readers understand the research question, methods used, and findings obtained. The authors can consider providing a brief outline or summary of the manuscript in the introduction section, which can help readers get a better understanding of the structure of the manuscript.

 ã€€

Response: We appreciate your suggestions. We have completely rewritten it under native English speakers, and we sincerely hope that the revised version will meet your approval.

(rewritten in brown characters)

Specific comments

=============

Major comments

---------------------

  1. The major consideration is that the authors needed to clarify the review part and the authors’ investigation part. 

 Response: We appreciate your guidance throughout the review article. We have made significant modifications and sincerely hope you will enjoy the new version.

Our original findings, however, have already been described in detail in two original articles, and citing them verbatim may conflict with the publication ethics issue of double submission. This review aims to provide a clinical picture based on our report, together with several previous reports that have already been published.  However, we appreciate your opinion that we should note our findings and have revised them to emphasise them.

  1. The authors should provide more information on the methods they used to conduct the review of existing literature. To do this, the authors can consider including a detailed description of the search strategy, including the databases used and the keywords searched.

Response: Thank you for your insightful comments. We have rewritten the entire document to incorporate your feedback and emphasise our uniqueness. As this is a review article, we cannot go into as much detail as possible in an experimental article, but we have summarized how we searched the literature. The specific course and Laboratory results of the cases we encountered have been published in two separate articles, so we have intentionally avoided duplicating them.

  1. Please clarify the content for Clostridium perfringens with/without MIH and other Clostridium pathogens. Please clarify the treatment for adult and newborn, especially for the treatment part.

Response: We appreciate your comments.  In this regard, we have revised how to use antibiotics to treat Clostridium perfringens with/without MIH and other Clostridium pathogens.

We are also aware that adult, adolescent, and newborn pathology varies. As there has not been a single case of C. perfringens infection in neonates reported in our hospital or previous studies, nor has there been a case report of a neonatal case resulting in MIH, we cannot mention the clinical features of the neonatal patients.

  1. Please add the age of non-MIH patients and statistical significance before the following sentences: “Therefore, our study shows that MIH patients are significantly younger than non-MIH patients.” Same as the following sentences: “In MIH patients, severe primary symptoms such as altered consciousness, severe pain, shock, haematuria, and gas formation occur more frequently than in those without

MIH.”

The subsequent was added for better understanding.

Clinical profiles of C.perfringens infection with/without MIH were summarized in Table 3

  1. Please add the prevalence of males to show more common.

 The subsequent was added.

( P6 L24-P7L4  ) Reported by us and others, MIH patients are suggested to be significantly younger than non-MIH patients. [2,3,5] Also it appears to be more prevalent in men, as reported previously 60%(13) and 58.1%(20) were males while the molecular basis of these gender differences is so far unknown.

  1. Please add the table to compare the background, serotype, clinical presentation, and laboratory data between C. perfringens with MIH and that without MIH from 106 cases.

Response: Thank you for highlighting the important point. Even though we have entered as much background, serotype, clinical picture, and laboratory data as possible in the table of 106 cases (Table 1), all reported cases are MIH cases, making comparisons with unaffected cases impossible.

Thank you for your comments. We have summarized the latest cases since 1991 in Table 2. We  are sorry that no detailed information was available before 1990

  1. Please add the poor prognosis as a number.

Thank you for the critical point. We have added the following.

 (13 L13-18) . Because reported cases suggested that life expectancy was significantly lower among MIH patients, with attributed mortality of 6/6, or 100%, compared to 13/54, or 24.1%, among non-MIH patients (p 0.001) The time between bacteraemia and death was 0.18 days (range: 0.04-1.08 days) in MIH versus 5.32 days (5-73 days) in controls (p 0.001).

Minor comments

---------------------

  1. Please clarify the legend in table 1. It was doubled.

 Response: Sorry, we have corrected it.

  1. Please elaborate on table 1.

 Response: Thanks. We have added several descriptions.

  1. Please add the arrow(s) in figure 1 to clarify the massive hemolysis for unfamiliar readers.

 Response: Thanks. We have added an arrow in Fig1.

Round 2

Reviewer 2 Report

General Comments:

=============

Thank you for the opportunity to review the manuscript on Clostridium perfringens bacteremia. Overall, the manuscript was well-elaborated, and the authors did an excellent job of presenting the findings. However, there were a few issues that need to be addressed, as outlined below.

Specific Comments:

=============

Major Comments:

---------------------

1. The authors mentioned that this is a review article. Therefore, it would be helpful if the authors could provide more information on the search strategy used to identify the approximately 100 cases included in the study. Specifically, the authors should clarify which databases were searched (e.g., Pubmed, MEDLINE, or others) and the search terms used to identify the cases.

2. In the sentence "The time between bacteremia and death was 0.18 days (range: 0.04-1.08 days) in MIH versus 5.32 days (5-73 days) in controls (p 0.001)", it would be helpful to clarify whether the time is the average, mean, or median time. This clarification will help readers to better understand the results.

Overall, the manuscript is well-written, and the authors did an excellent job of presenting the findings. However, addressing the issues mentioned above will help to improve the manuscript's clarity and readability.

Author Response

Response letter

Thank you for the opportunity to review the manuscript on Clostridium perfringens bacteremia. Overall, the manuscript was well-elaborated, and the authors did an excellent job of presenting the findings. However, there were a few issues that need to be addressed, as outlined below.

Response:

Our deepest thanks to you, too, for your careful peer review. I am delighted that you have so highly valued the revised manuscript. I have made more revisions to your points, and I hope you will be satisfied with them.

Specific Comments:

=============

Major Comments:

---------------------

  1. The authors mentioned that this is a review article. Therefore, it would be helpful if the authors could provide more information on the search strategy used to identify the approximately 100 cases included in the study. Specifically, the authors should clarify which databases were searched (e.g., Pubmed, MEDLINE, or others) and the search terms used to identify the cases.

Response: We have added the followings for a more accurate presentation of our research methods.

P4L10-15  To clarify the clinical and bacteriological profiles of MIH caused by C. perfringens bacteraemia, a systematic search of Pubmed over the past seven decades was conducted in this study. We searched for every English-language case report with C. perfringens and hemolysis(haemolysis) in the title, abstract, or main body. Then, we compared our findings to those of earlier studies.

  1. In the sentence "The time between bacteremia and death was 0.18 days (range: 0.04-1.08 days) in MIH versus 5.32 days (5-73 days) in controls (p 0.001)", it would be helpful to clarify whether the time is the average, mean, or median time. This clarification will help readers to better understand the results.

We really appreciate your comment. This is our mistake. That is not the average but the median time. To avoid misunderstanding, we have revised it as follows.

P13 L18-20

The mean time between bacteraemia and death was 0.18 days (range: 0.04-1.08 days) in MIH versus 32 days (5-73 days) in controls (p 0.001).

Overall, the manuscript is well-written, and the authors did an excellent job of presenting the findings. However, addressing the issues mentioned above will help to improve the manuscript's clarity and readability.

Response;

Again, thank you very much for your kind peer review!

P/S

We have also made some revisions to typo errors and deleted the following sentences to avoid redundancy.

P2L7

  1. perfringens is classified into seven types, A, B, C, D, E, F and G.

P14L17-20

Antibiotic therapy alone is insufficient for treating clostridial infections. Thus, antitoxin therapy has already been established for infections brought on by C. tetani and C. botulinum